# Animal Origin Bioactive Hydroxyapatite Thin Films Synthesized by RF-Magnetron Sputtering on 3D Printed Cranial Implants

**Diana Chioibasu [1,2], Liviu Duta [1], Gianina Popescu-Pelin [1], Nicoleta Popa [3], Nichita Milodin [3], Stefana Orobeti (Iosub) [1,4], Liliana Marinela Balescu [5], Aurelian Catalin Galca [5], Adrian Claudiu Popa [5,6], Faik N. Oktar [7,8], George E. Stan [5,* ] and Andrei C. Popescu [1,*]**

[1] National Institute for Lasers, Plasma and Radiation Physics, RO-077125 Magurele-Ilfov, Romania; diana.chioibasu@inflpr.ro (D.C.); liviu.duta@inflpr.ro (L.D.); gianina.popescu@inflpr.ro (G.P.P.); stefana.iosub@inflpr.ro (S.O.)

[2] Faculty of Applied Sciences, Department of Physics, University Politehnica of Bucharest, RO-060042 Bucharest, Romania

[3] National Institute for Mechatronics & Measurement Technique, RO-021631 Bucharest, Romania; nicoleta.popa@incdmtm.ro (N.P.); nichita.milodin@incdmtm.ro (N.M.)

[4] Department of Molecular Cell Biology, Institute of Biochemistry of the Romanian Academy, RO-060031 Bucharest, Romania

[5] National Institute for Materials Physics, RO-077125 Magurele-Ilfo, Romania; liliana.trinca@infim.ro (L.M.B.); ac_galca@infim.ro (A.C.G.); adrian.popa@gmail.com (A.C.P.)

[6] Army Centre for Medical Research, Bucharest RO-010195, Romania

[7] Center for Nanotechnology & Biomaterials Research, University of Marmara, Goztepe Campus, 34722 Istanbul, Turkey; foktar@marmara.edu.tr

[8] Department of Bioengineering, Faculty of Engineering, University of Marmara, Goztepe Campus, 34722 Istanbul, Turkey

**\*** Correspondence: andrei.popescu@inflpr.ro (A.P.); george_stan@infim.ro (G.E.S.); Tel.: +40-21-457-4550 (A.P.); +40-21-241-8128 (G.E.S.)

**Abstract:** Ti6Al4V cranial prostheses in the form of patterned meshes were 3D printed by selective laser melting in an argon environment; using a $CO_2$ laser source and micron-sized Ti6Al4V powder as the starting material. The size and shape of prostheses were chosen based on actual computer tomography images of patient skull fractures supplied in the framework of a collaboration with a neurosurgery clinic. After optimizations of scanning speed and laser parameters, the printed material was defect-free (as shown by metallographic analyses) and chemically homogeneous, without elemental segregation or depletion. The prostheses were coated by radio-frequency magnetron sputtering (RF-MS) with a bioactive thin layer of hydroxyapatite using a bioceramic powder derived from biogenic resources (Bio-HA). Initially amorphous, the films were converted to fully-crystalline form by applying a post-deposition thermal-treatment at 500 °C/1 h in air. The X-ray diffraction structural investigations indicated the phase purity of the deposited films composed solely of a hexagonal hydroxyapatite-like compound. On the other hand, the Fourier transform infrared spectroscopic investigations revealed that the biological carbonatation of the bone mineral phase was well-replicated in the case of crystallized Bio-HA RF-MS implant coatings. The in vitro acellular assays, performed in both the fully inorganic Kokubo's simulated body fluid and the biomimetic organic–inorganic McCoy's 5A cell culture medium up to 21 days, emphasized both the good resistance to degradation and the biomineralization capacity of the films. Further in vitro tests conducted in SaOs-2 osteoblast-like cells showed a positive proliferation rate on the Bio-HA RF-MS coating along with a good adhesion developed on the biomaterial surface by elongated membrane protrusions.

**Keywords:** biological origin hydroxyapatite; bioactive layers; cranial mesh implants; selective laser melting; 3D printing; radio-frequency magnetron sputtering

## 1. Introduction

Ti6Al4V is the most popular titanium alloy used in biomedical applications due to its high strength, high fracture toughness, superior corrosion resistance, low density, excellent biocompatibility, and favorable osseointegration [1–3]. Even if Ti6Al4V is considered as a highly biocompatible metallic material, there have been reports of inflammatory reactions occurring around the prostheses upon implantation [4–8]. In order to improve the biocompatibility of the metallic alloys, a bioactive material can be deposited on its surface as a thin film, in order to increase the osseointegration and to ensure the biomimetism of the implant [9–15]. Hydroxyapatite (HA), with the theoretical chemical formula $Ca_{10}(PO_4)_6(OH)_2$, in its non-stoichiometric multi-cation and anion substituted form, represents the main constituent of the mineral part of the bone. By the application of such a biomimetic coating, one can stimulate the osteoblasts to proliferate and to generate osseous matrix, which will culminate with the new bone formation. As stated, biological apatite (further denoted Bio-HA) is similar, but not chemically and structurally identical with the pure synthetic HA. Bio-HA can enclose a series of trace elements (e.g., Na, Mg, Sr, Zn, F), as well as substituting functional groups such as carbonate ($CO_3^{2-}$), silicate ($SiO_4^{3-}$), and/or hydrogen phosphate ($HPO_4^{-}$) [16–19]. Thereby, an apatite compound similar to the one produced in the bones of living mammals is very difficult, if not impossible, to replicate by chemical synthesis. This is why the extraction of Bio-HA from animal bones could represent a viable solution, not only due to the intrinsic natural doping with ions which can promote specific biofunctional traits, but also due to their high availability and reduced costs [17,20]. For biomedical research purposes, bovine [21,22], ovine [23], or swine bones [24] are the most commonly used resources, but also fish [25] and sea-shells [26], as well as more exotic sources such as camel [27] or shark [28] have been explored.

For depositing calcium phosphate-based coatings onto metallic implants, the industrial technique of choice is plasma spray (PS), due to the deposition speed, large area coverage, and inexpensive working atmosphere (i.e., ambient or argon) [29,30]. However, HA coatings produced using this technique could contain residual decomposition phases because of high-processing temperatures, having different dissolution speeds, and thereby an unpredictable biological behavior [31]. Pulsed laser deposition (PLD) is another coating technique which proved reliable for the fabrication of bioactive layers [6,17,32]. The PLD strength comes from its capability to congruently transfer from target-to-substrate even materials with complex stoichiometry, but the biggest shortcoming is the limitation to a very small deposition area of the order of few $cm^2$ [32]. Radio-frequency magnetron sputtering (RF-MS) was long-time considered a solid alternative to PS since it can produce (by the control of deposition parameters) dense, pure, highly-adherent films on large area substrates (even on ones with complex geometry when coupled with rotation-translation facilities) [33–35]. There are reports in scientific literature on deposition of synthetic pure [7,12,34] or substituted HA [11,13,36] coatings using RF-MS. However, to the best of our knowledge, no information has been reported yet on the successful magnetron sputtering of Bio-HA films and consequently on their structural and biofunctional properties.

Casting and forging are the conventional processing technologies used for implants mass manufacturing. When customized prostheses with dimensions and complex geometries are in demand, such technologies are not reliable because of the time, energy, material consumption, and costs [37]. As an alternative, laser-based additive manufacturing (AM) is a method that allows customized, cost-effective, and rapid production of complex 3D medical parts [38,39].

Selective laser melting (SLM) is an AM process capable of producing metallic, ceramic, plastic, or composite parts using layer-by-layer manufacturing [40–43]. 3D structures are generated from a geometric CAD (computer aided design) file. The model is virtually sliced into 2D contour lines and

imported into the AM machine that builds the new part [44–47]. The energy source is a laser beam which scans a selected area of a powder bed. The area scanned by the laser beam is molten and it quickly solidifies. The new solid structure is covered by a powder layer with a leveling system, and subsequently the laser scanner irradiates the next contour and thus, layer-by-layer, the part is completed [48,49].

SLM has been used to manufacture hip, ankle, knee, tibia, maxillo-facial, or spinal prostheses with intricate geometrical parts, based on medical imaging. The correlation between micro-structural features and mechanical properties has been reported in the literature [37,50–53]. Fatigue strength of AM components is the primary mechanism of failure, and for this reason many investigations have been conducted in this area of research [54,55]. Lieverani et al. [56] reported the fabrication of Co-Cr-Mo endoprosthetic ankle devices using SLM. They identified the optimal process parameters for obtaining samples with maximum density and high mechanical strength.

We report in this study the manufacturing in two steps of a cranial prosthesis with bioactive Bio-HA surface with dimensions corresponding to a cranial large defect of a real patient, obtained by computer tomography. First, the prosthesis is generated by 3D printing using the SLM method, while in a second step, a thin film is deposited by RF-MS from a target of biological apatite derived from bovine bones. The focus of this study was to assess for the first time the feasibility of depositing by RF-MS bioceramic coatings of biogenic origin onto SLM custom-fabricated prosthesis. In this respect, morphological, compositional, structural, and biofunctional interrogations were conducted for both the metallic part and the bioceramic coating.

## 2. Materials and Methods

### 2.1. Cranial Meshes

#### 2.1.1. 3D Printing of Cranial Meshes

For the manufacturing of meshes, a Ti6Al4V powder with the particle size in the range of 10–30 μm (Eos GmbH, Krailling, Germany) was used. The SLM machine was a model M270 (Eos GmbH, Krailling, Germany).

The laser beam used for melting the metallic powder was generated by a $CO_2$ laser source ($\lambda$ = 10.6 μm, continuous wave, Gaussian beam) connected to a scanner. The optimal laser parameters recommended by the producer for SLM of titanium alloys are: 90 W laser power and 450 mm/s scanning speed, and an under 4 bar argon atmosphere. The laser spot on the powder bed was of ~100 μm. For the manufacturing of a 3 mm thick cranial mesh prosthesis a number of 100 successive layers were needed, corresponding to an individual layer thickness of 30 μm.

The cranial meshes were designed in the SolidWorks (Dassault Systems, Vélizy-Villacoublay, France) graphical engineering software, and imported in the EOSPRINT proprietary software of the SLM machine, which generated the movement of the scanner mirrors.

The prostheses were extracted from the Ti plate by electro-erosion cutting using a Robofil 6030 machine (Charmilles Technologies, Schaffhausen, Switzerland) with 0.25 mm thickness of the wire, 13 m/min wire consumption, and a cutting speed of 1 mm/min. Prostheses of 3 mm thickness were produced by SLM, so that after cutting, the thickness of the mesh to be of ~2 mm.

#### 2.1.2. Characterization of the 3D Printed Material

First, bulk prisms of $10 \times 10 \times 20$ mm$^3$ were 3D printed in view of metallographic characterization, in the same experimental conditions as the meshes. The obtained samples were cut in slices using a disk cutting machine, model Brillant 200 (ATM GmbH, Mammelzen, Germany). The slices were fixed in Bakelite using a machine Opal 410 (ATM GmbH, Mammelzen, Germany) and polished with a Saphir 520 (ATM GmbH, Mammelzen, Germany) semiautomatic machine. The resulting mirror-polished samples were etched with Kroll solution (water 92%, nitric acid 6%, hydrofluoric acid 2%) in order to reveal the metallographic microstructure of the alloy after SLM.

The metallographic analysis was conducted with a BX51M (Olympus, Tokyo, Japan) optical microscope and a scanning electron microscope (SEM) Quanta 450 (FEI Company, Hillsboro, OR, USA). The compositional homogeneity of the prosthesis has been investigated by energy dispersive X-ray spectroscopy (EDXS) measurements.

The micro-hardness of the printed material was evaluated by the Falcon 500 mechanical testing machine (Innovatest, Maastricht, Netherlands) equipped with a Vickers indenter. Ten Vickers indentations have been made on the surface of the specimens using a load of 0.5 kgf/mm$^2$. The reported micro-hardness value represents the average of the ten independent values ± standard deviation.

*2.2. Coating of the Prosthesis with a Bio-HA Thin Film*

The final step of the manufacturing process was to coat the 3D printed prosthesis with a thin ceramic layer of biological hydroxyapatite (Bio-HA) extracted from bovine bone for increasing the bioactivity of the surface.

2.2.1. Bio-HA Extraction

The extraction of Bio-HA was performed by applying a previously established work protocol [17,23,57], conducted in accordance with the European Regulation 722/2012 and ISO 22442-1:2015—Medical devices using animal tissues and their derivatives—Part 1: Application of risk management. In brief, the bone marrow and the soft tissue residues were removed mechanically from bovine femoral bodies. The femoral shafts were cut into small pieces and ultrasonicated with distilled water. The cleaned parts were deproteinized by immersion for 14 days in an alkali water solution (1% sodium hypochlorite). Next, they were washed with water and dried, and then submitted to a calcination at 850 °C for 4 h in air. In the end, the calcined (white-colored) bone specimens were ball-milled into fine Bio-HA powders.

2.2.2. Deposition of Bio-HA Thin Films by RF-MS

Ti6Al4V cranial meshes manufactured by SLM were used as substrates together with monocrystalline infrared transparent Si <111> wafers (Medapteh, Magurele, Romania). The later ones were required to enable the structural evaluation of deposited films by Fourier transform infrared (FTIR) spectroscopy in transmission mode.

The substrates were first degreased in acetone and further on cleaned in iso-propanol using an ultrasonic bath and a dwell time of 15 min per step. In the end, they were fast-dried by argon gas purging and mounted on a rotating stage of the deposition chamber of the UVN-75R1 (Vacma, Kazan, Republic of Tatarstan, Russian Federation) RF-MS equipment.

The target (with the diameter of 110 mm) has been manufactured by mild-pressing the Bio-HA powder at room-temperature (RT) in a specially designed Ti plate (crucible). Such type of targets are preferred to the compacted ceramic disks (prone to cracking during prolonged sputtering processes), as they provide not only consistency, but also reducing significantly the fabrication costs [58,59].

First, a base pressure of ~3 × 10$^{-3}$ Pa was achieved in the working chamber, such as to ensure a low contamination level and thus enable the fabrication of good purity films.

Prior to deposition, the substrate and the target were subjected individually to plasma cleaning treatments. The substrates were cleaned in situ for 15 min by argon ion etching following the protocol described in [60]. The target was pre-sputtered in the future deposition conditions for 30 min. This step is essential since it can ensure the removal of contaminants adsorbed on the target surface, and at the same time, it can allow the stabilization of the sputtering processes before the actual deposition [59,61].

The Bio-HA thin film deposition was performed at an argon pressure of 0.3 Pa, without intentional substrate heating (the substrate reached a temperature of ~120 °C owned to radiative plasma bombardment processes only). A 1.78 MHz RF generator and an electrical power of ~100 W were used. The target-to-substrate distance was set at 35 mm. Based on the deposition rate of ~3.7

nm/min (previously estimated by spectroscopic ellipsometry (SE) measurements), Bio-HA films of ~600 nm thickness were deposited. In order to crystallize the initially amorphous as-deposited Bio-HA films, without inducing a strong oxidation of the Ti6Al4V implant substrate, a post-deposition thermal-treatment was applied at 500 °C/1 h in air.

*2.3. Characterization of Biological HA Thin Films*

2.3.1. Physico–Chemical Characterizations

Since the RF-MS technology produces smooth layers [33,62] and the SEM method does not have enough contrast to distinguish the fine topological details, the morphology of Bio-HA films deposited onto mirror-polished substrates could be better revealed by atomic force microscopy (AFM) analysis. An NTEGRA Probe NanoLaboratory System (NT-MDT, Moscow, Russian Federation) equipped with an NSG01 cantilever with 10 nm tip was used. The AFM images were recorded in non-contact mode on areas of 25 × 25, 10 × 10, 5 × 5, and 2 × 2 $\mu m^2$.

The comparative composition of source material and Bio-HA RF-MS deposited film was evaluated by EDXS using an EDAX (Ametek, Berwyn, PA, USA) equipment with a Si-Li detector, operated at 20 kV acceleration voltage. The EDXS measurements were performed in at least four regions with an area of ~240 × 240 $\mu m^2$, randomly chosen on the sample surface. For the calibration of Ca/P atomic ratio, a National Institute of Standards and Technology standard reference sample of HA (i.e., NIST-SRM 2910b) was measured as well. Arithmetic means and standard deviations (SD) have been inferred, and statistical analyses were carried out using the unpaired two-tailed Student's t-test, with the differences being considered significant when $p < 0.01$.

The evaluation of the crystalline status of source powder and Bio-HA films was performed by X-ray diffraction (XRD), in the angular range $2\theta = 9$–$54°$, using a step of $0.04°$. The source material and the heat-treated Bio-HA film were investigated comparatively using symmetrical geometry (Bragg–Brentano) XRD measurements, using a D8 advance diffractometer (Bruker, Karlsruhe, Germany) with CuK$_{\alpha 1}$ radiation. In addition, the films were measured, before and after the post-deposition thermal-treatment, also by grazing incidence XRD (GIXRD) at an incidence angle of $0.5°$, using a SmartLab 3 kW diffractometer (Rigaku, Tokyo, Japan), in order to maximize their diffracted signal.

The chemical structure of the source powder and deposited Bio-HA films, with a focus on the short-range order and the transfer of functional groups specific to the bone mineral component, was investigated by Fourier transform infrared (FTIR) spectroscopy, in transmission mode. A Spectrum BX II (Perkin Elmer, Waltham, MA, USA) apparatus was used. The spectra were recorded in the wave numbers range of 400–4000 cm$^{-1}$, at a resolution of 4 cm$^{-1}$, and represent the average of 32 individual scans per sample.

2.3.2. In Vitro Tests in Simulated Physiological Solutions

An implant coating has to play a double role: (i) act as a buffer against the release of metallic ions of the metallic substrate in the internal environment and (ii) promote osteoconductivity. In vitro tests in synthetic physiological fluids were performed in order to evaluate the resistance to degradation of the deposited coatings and to explore their biomineralization capability. The tests were carried out following the indications contained in the ISO 23317:2014 protocol: "Implants for surgery—In vitro evaluation for apatite-forming ability of implant material". However, besides the fully inorganic Kokubo's simulated body fluid (SBF), recommended by the ISO standard, we have also used for these tests the complex biomimetic inorganic–organic McCoy's 5A culture medium supplemented with 15% fetal bovine serum (FBS)—the same medium formulation used for the in vitro tests on cell cultures. The immersed samples were kept at 37 °C in a stabilized biology incubator in humid atmosphere (to avoid the medium evaporation) under the accurate homeostatic conditions (5% $CO_2$) up to 21 days (we note that the ISO 23317:2014 does not provide recommendations for the testing atmosphere, and the implied common use of atmospheric pressure seems faulty). After 7 and

21 days, the Bio-HA films were extracted, gently washed with distilled water, and left to dry in a desiccator. Subsequently, they were investigated by SE, AFM, and FTIR spectroscopy.

The SE measurements were performed using a spectroscopic ellipsometer (J.A. Woollam Co., Lincoln, NE, USA) equipped with a HS-190 monochromator and a goniometer that allows for automatic acquisition of spectra at different angles of incidence. For Bio-HA films before and after immersion in biological medium, the measurements were performed at 65°, 70°, and 75° incidence angles in the spectral range of 1.3–3.5 eV. The Bio-HA material optical properties (refractive index dispersion) were inferred using a Cauchy formula [63].

### 2.3.3. In Vitro Tests on Cell Cultures

SLM printed prism samples were cut into $10 \times 10$ mm$^2$ coupons to serve as substrates for in vitro cell biocompatibility tests. The SLM coupons were polished down to mirror level to exclude any possible morphological incongruences between specimens, and their influence on cell behavior. The Bio-HA coated samples were subjected to sterilization in water vapors at 121 °C for 30 min using an AES-8 autoclave (Raypa, Barcelona, Spain).

Experiments were performed with human osteosarcoma (SaOs-2) cells which were grown in McCoy's 5A medium (Gibco Thermo Fisher Scientific, Waltham, MA, USA) supplemented with 15% FBS (Gibco Thermo Fisher Scientific, Waltham, MA, USA), 50 U/mL penicillin and 50 mg/mL streptomycin (Gibco Thermo Fisher Scientific, Waltham, MA, USA). Cells were seeded on biomaterials at a density of 2000 cells/cm$^2$. All substrates were then kept in a humid atmosphere with 5% $CO_2$ at 37 °C for 1, 3, and 7 days, respectively. Borosilicate glass coverslips were used as experimental standard control material.

Cell proliferation was assessed by using a CellTiter 96® AQueous one solution cell proliferation assay (MTS) kit (Promega, Madison, WI, USA). For this, samples were cultured in triplicate for 1, 3, and 7 days. At each time interval, samples were transferred to new wells and incubated at 37 °C for 1 h 30 min with fresh medium containing the MTS reagent. The absorbance values were recorded at 450 nm using a LB 913 Apollo 11 spectrophotometer (Berthold Technologies, Bad Wildbad, Germany). The final data was represented as the average of the measured optical density upon background subtraction. Statistical analysis was performed by using GraphPad Prism 6 (GraphPad, San Diego, CA, USA) in which an unpaired, two-tailed, Student's t-test was calculated. Differences were considered statistically significant for $p < 0.05$.

Cell adhesion was examined by immunofluorescence microscopy. After 24 and 72 h of culture, SaOs-2 cells were fixed for 15 min with 4% paraformaldehyde at RT and stored in phosphate buffered saline solution (PBS) at 4 °C prior to labelling. Fixed cells were then permeabilized for 3 min with 0.2% TritonX-100 and blocked for 1 h in 0.5% bovine serum albumin (BSA). To visualize the actin filaments, cells were labelled with 1:100 diluted Alexa Fluor 488 conjugated Phalloidin (Cell Signaling Technology, Danvers, MA, USA). Cells were subsequently treated with 1 µg/mL Hoechst (Cell Signaling Technology, Danvers, MA, USA) to stain cell nuclei. Upon each incubation, samples were washed three times with PBS. Finally, representative images were acquired on each substrate by using a Leica DM 4000 B LED fluorescence microscope equipped with a DFC 450 C camera (Leica Microsystems GmbH, Wetzlar, Germany) and the appropriate filters.

## 3. Results

### 3.1. Printing of Cranial Mesh Prostheses and Their Structural and Chemical Characterization

A photograph of a cranial mesh grown by SLM is presented in Figure 1. The pattern design is original, being devised after studying the morphology of several commercial cranial prostheses. Structures of $50 \times 50 \times 3$ mm$^3$ with the pattern depicted in Figure 1b were built layer-by-layer on a 50 mm thick Ti plate by laser scanning. Every single contour was ~30 µm tall and 100 layers were needed to obtain a final product with the desired height.

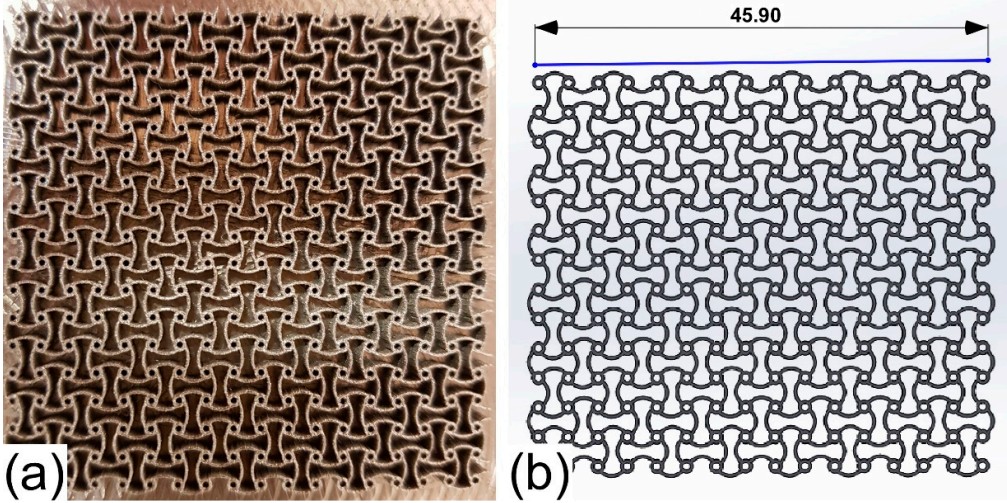

**Figure 1.** (**a**) Photograph of a Ti6Al4V cranial mesh manufactured by SLM (Selective laser melting) based on a (**b**) three-dimensional (3D) CAD (computer aided design) model generated in the SolidWorks Professional software, version 2013 (Dassault Systems, Vélizy-Villacoublay, France).

After polishing and attacking with a metallographic reagent, the grain structure was exposed and it was studied by optical and SEM microscopies (Figure S1 of Supplemental Information). Optical and SEM micrographs, characteristic to the printed SLM printed Ti6Al4V sample are presented in Figure S1a,b, respectively. Figure S1a displays a martensitic acicular structure that developed in islands in the $\alpha$ phase. More specifically, it is an $\alpha'$-type martensite that crystallizes in the hexagonal system upon the fast cooling of biphasic titanium alloys ($\alpha + \beta$) such as Ti6Al4V. The acicular plates displayed a "clustered" morphology, consisting of structural units with needles exhibiting a certain parallelism. This distribution includes the microstructure of the printed material in the cryptocrystalline martensite category. In general, $\alpha'$ martensite nucleates at the boundaries of the prior $\beta$ grains and spreads within their parent $\beta$ grains, which explains the different orientation of the $\alpha'$ needles. SEM investigations allowed for a better visualization of the grains, the parallel distribution of lamellar martensite was more obvious, with varying lengths of the grains dependent on the micro-regions from which they were part of (Figure S1b). There are structural units with grains whose average lengths are greater than 100 µm, which correspond to primary $\alpha'$ martensite, but also units with finer needles, with lengths of about 15–20 µm, corresponding to secondary $\alpha'$ phase. These later grains were perpendicular to the primary $\alpha'$ ones.

The hardness tests performed in 10 randomly chosen points of the SLM test sample revealed similar hardness values, which represent an indirect proof of its structural homogeneity. The average hardness of the SLM prints was 391 ± 5 HV, similar to the casted Ti6Al4V alloy [64].

Subsequently, the same sample was analyzed by EDXS. An EDXS characteristic spectrum of a Ti6Al4V 3D printed sample is shown in Figure S2a (of Supplemental Information). The elemental composition was found to be ~90.2 wt% Ti, ~5.6 wt% Al, and ~4.2 wt% V, thus, congruent to the one of the Ti6Al4V starting powder.

Further, the EDXS mapping analysis (Figure S2b–e) attested the presence of Ti, Al, and V, the components of the studied alloy (Figure S2b), and offered qualitative information on the uniform distribution of elements (Figure S2c–e). The brightness is directly proportional to the presence of the element being tracked. Taking into account this principle, it was confirmed that all the components were uniformly distributed over the studied area, with no trace of elemental segregation or depletion.

### 3.2. Functionalization of the Cranial Mesh with a Bioceramic Layer

Illustrative images of a Ti6Al4V cranial mesh, before and after biofunctionalization with a Bio-HA thin film by RF-MS, are presented in Figure 2.

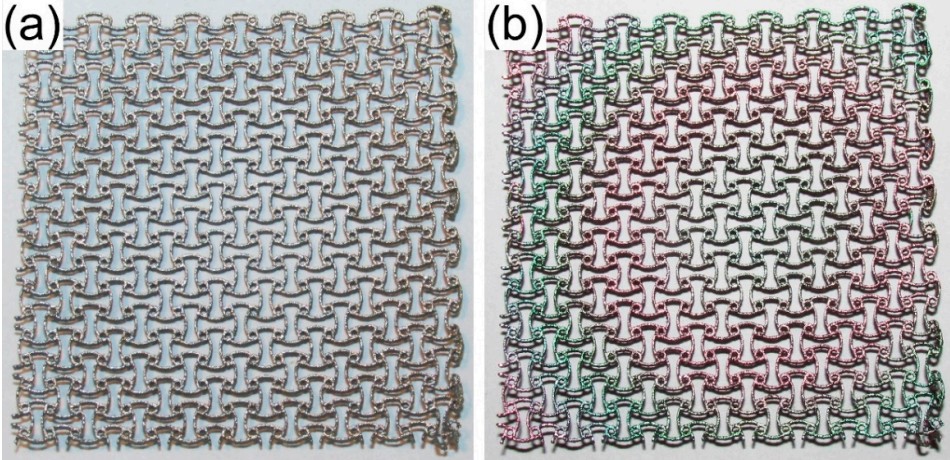

**Figure 2.** (**a**) Uncoated and (**b**) Bio-HA (derived from biological hydroxyapatite) sputtered cranial mesh of Ti6Al4V fabricated by SLM.

### 3.2.1. Physico–Chemical Characterization of the Bio-HA Functional Layer

AFM micrographs, collected at different magnifications, are depicted in Figure 3. They highlighted that the crystallized Bio-HA RF-MS film consisted of a compact matrix of polyhedral grains with diameters in the range ~110–230 nm. An average root mean square roughness ($R_{rms}$) of ~15 nm was inferred, irrespective of the scanned microscopic field area, which advocates for the good uniformity of the deposited film.

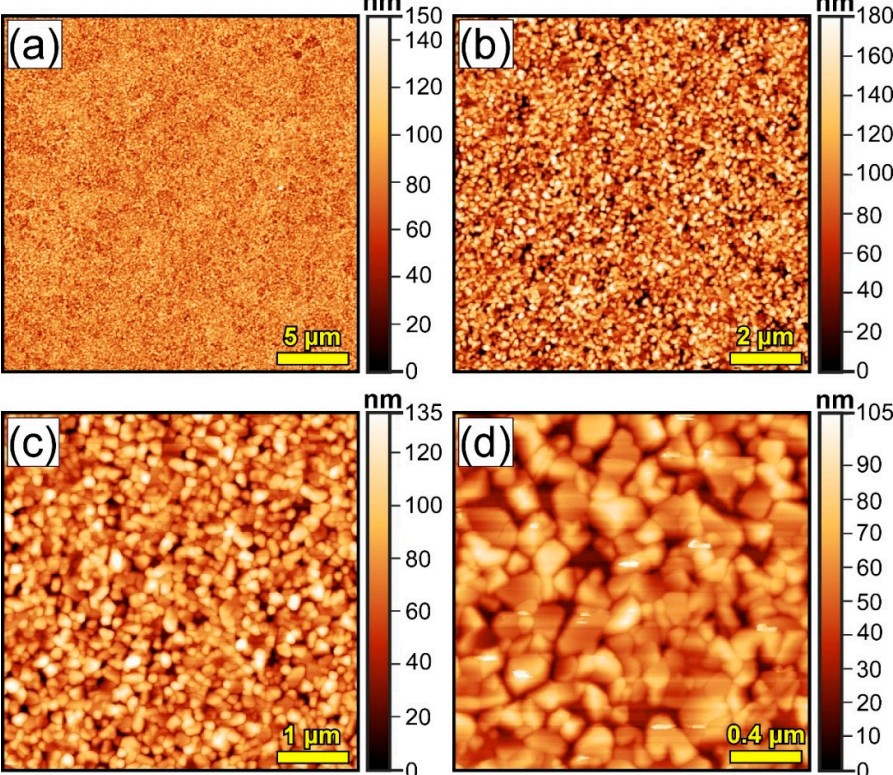

**Figure 3.** Characteristic AFM (atomic force microscopy) images of the of Bio-HA films surface acquired on scanning areas of: (**a**) 25 × 25; (**b**) 10 × 10; (**c**) 5 × 5; and (**d**) 2 × 2 μm², respectively.

Next, the composition of the Bio-HA sputtered film and parent target were determined by EDXS and compared. In addition, the Ca, P, and O, also small contents of Na (~3 at%) and Mg (~2 at%) were detected. Such specific trace elements present in the composition of bone apatite, are known to play important roles in bone metabolism and health [65,66], and thereby, their successful transfer by RF-MS into the deposited layers can be emphasized. The mean elemental concentration ± standard deviations obtained for each of these constituent elements (except oxygen, light element for which the EDXS quantification yields characteristically large errors) of the Bio-HA target and film are shown in Figure 4. Significant compositional differences ($p < 0.01$) were found only in the case of P and Mg. The Ca/P molar ratios for the source material and the Bio-HA film were of 1.63 ± 0.02 and 1.70 ± 0.01, respectively. However, in the case of cation substituted HAs, such as those derived from biogenic resources, the contribution of natural dopants (e.g., Na, Mg) should not be dismissed. In this case, a (Ca + Na + Mg)/P molar ratio could provide further information on the features of the ceramic compounds. The (Ca + Na + Mg)/P molar ratios of the Bio-HA target and film were of 1.76 ± 0.01 and 1.85 ± 0.01, respectively (thus, situated close to the Ca/P ratio of 1.67 of stoichiometric HA, $Ca_{10}(PO_4)_6OH_2$). The higher (Ca + Na + Mg)/P ratios (than 1.67), obtained in the case of both target and sputtered film, are suggesting a deficit of orthophosphate groups, most probably due to their substitution with carbonate functional groups (specific to Bio-HAs). A slight increase of the (Ca + Na + Mg)/P was recorded in the case of the RF-MS film with respect to the parent target. This is connected with the P evaporation phenomena taking place during sputtering deposition [34,59].

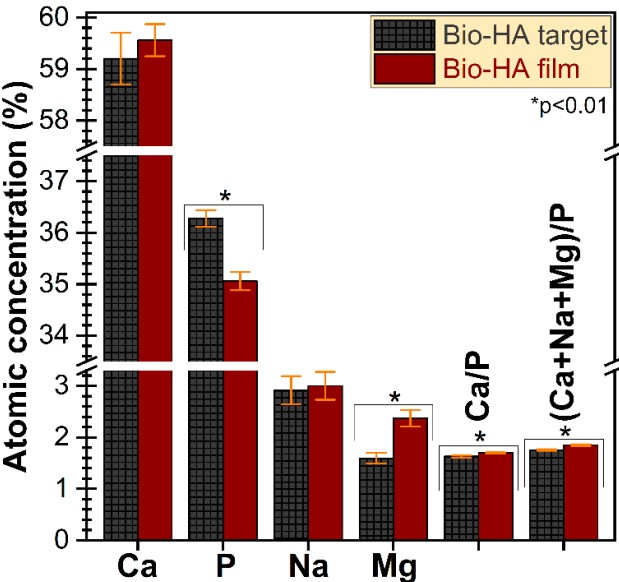

**Figure 4.** Elemental composition of the Bio-HA film produced by RF-MS (radio-frequency magnetron sputtering) with respect to the source target material. The Ca/P and (Ca + Na + Mg)/P molar ratios of the Bio-HA target and film.

XRD patterns were recorded in symmetric geometry (Figure 5a,b), for both Bio-HA target and crystallized Bio-HA film, to accommodate an equivalent structural comparison and to survey the presence of preferred crystallite orientations (if any). It has been revealed that the HA powder obtained from the bovine cortical bone was monophasic, consisting of a randomly oriented hexagonal hydroxyapatite compound (ICDD-PDF4: 00-009-0432) only. In the case of the thermal-treated (TT) Bio-HA film, besides the intense diffraction peaks of the Ti6Al4V substrate (crystallized in the hexagonal system, ICDD-PDF4: 04-020-7055), only peaks of a well-defined phase of hexagonal HA were evidenced. The increased intensity of the 002 peak of Bio-HA film with respect to the ICDD file indicated that the Bio-HA crystallites elicited a preferred orientation along the *c*-axis, normal to the substrate. The *c*-axis texturing is characteristic to sputtered films synthesized from materials of the

hexagonal system [34,36,60,62,67]. The average crystallite size determined based on the full width at the half-maximum (FWHM) of (002) planes reflections was estimated by applying the Scherrer equation [68]. The FWHM was corrected for instrumental breadth using a corundum NIST-SRM 1976a sample. Similar average crystallite sizes of ~167 and ~139 nm were obtained in the case of the source target material and crystalline Bio-HA film, respectively. This finding demonstrated the capability of the chosen thermal treatment to yield excellent crystalline quality. Furthermore, the efficiency of post-deposition thermal treatment performed in air at 500 °C/1 h was highlighted by GIXRD measurements (Figure 5c,d) which showed the complete conversion of the initial amorphous as-sputtered film (see the halo centered at $2\theta \approx 30°$) into a completely crystallized Bio-HA structure.

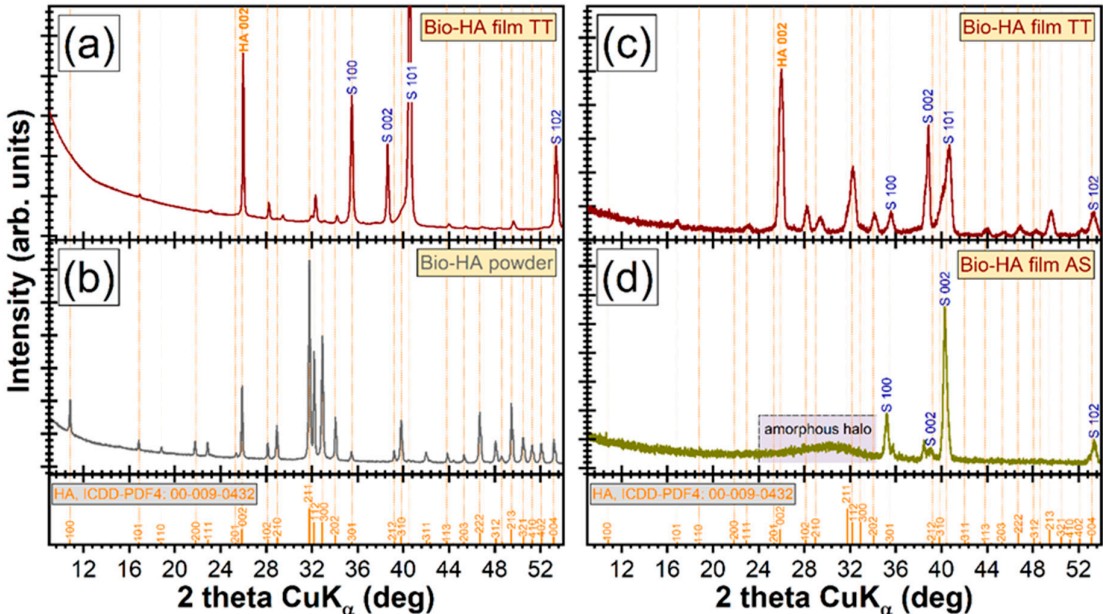

**Figure 5.** XRD (X-ray diffraction) patterns recorded in (**a,b**) symmetric (Bragg–Brentano) and (**b,d**) grazing incidence ($\alpha$ = 0.5°) geometries in the case of (**b**) Bio-HA source material and (**d**) as-sputtered (AS) and (**a,c**) thermal-treated (TT) Bio-HA film. For comparison, the ICDD-PDF4: 00-009-0432 reference file of pure hexagonal hydroxyapatite is inserted on the bottom of the figures.

The short-range order of the Bio-HA magnetron sputtered film and the presence and type of functional groups incorporated within its structure was surveyed by FTIR spectroscopy measurements (Figure 6). The sharp and well-split IR absorption bands of orthophosphate groups recorded in the case of the Bio-HA film (Figure 6a) testified for it its high crystallinity [69], in full agreement with the XRD data. All the characteristic IR bands of HA were evidenced for the Bio-HA sputtered structure: $\nu_4$ bending (~565 and ~600 cm$^{-1}$), $\nu_1$ symmetrical stretching (~962 cm$^{-1}$), and $\nu_3$ asymmetric stretching (~1032 and ~1088 cm$^{-1}$) vibration modes of orthophosphate groups, and $\nu_L$ libration (~632 cm$^{-1}$) and $\nu_S$ stretching (~3753 cm$^{-1}$) of structural hydroxyl units [26,36,69]. Only slight shifts and intensity modification of the Bio-HA film bands (Figure 6a) with respect to the pure HA reference and Bio-HA source powder (Figure 6b) were noticed. These are denoting minor structural alterations (in terms of bond length and/or angle and short-range order), and are to be expected for a material deposited by magnetron sputtering, thus, under non-equilibrium thermodynamic conditions. Furthermore, it can be emphasized the successful incorporation of carbonate groups in the orthophosphate sites (B-type carbonatation) of the Bio-HA film crystalline lattice, similarly to the source powder: $\nu_2$ symmetrical (~875 cm$^{-1}$) and $\nu_3$ asymmetrical (~1417 and ~1472 cm$^{-1}$) stretching vibrations of $(CO_3)^{2-}$ units [69]. Such type of substitution is also characteristic to the mineral phase of the human bone [36,70].

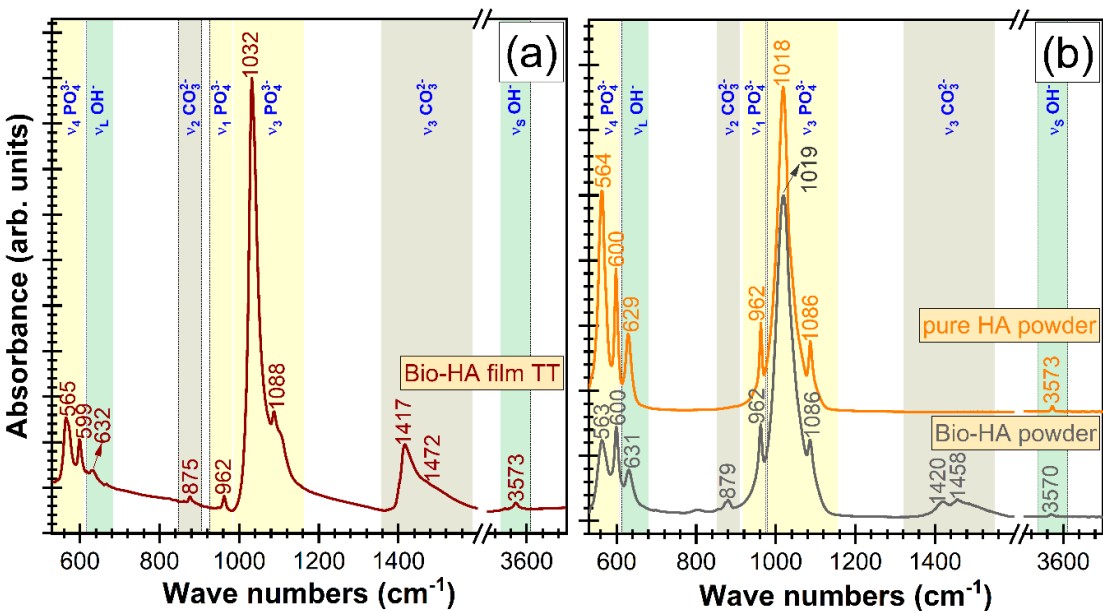

**Figure 6.** Comparative FTIR (Fourier transform infrared) spectra of (**a**) Bio-HA crystallized film and (**b**) pure commercial HA powder (Acros Organics, Geel, Belgium) and Bio-HA source material.

### 3.2.2. In Vitro Tests in Simulated Physiological Solutions

The first step in the evaluation of in vitro behavior of Bio-HA coatings was to immerse them in both a fully inorganic SBF solution (as the ISO 23317/2014 standard recommends), as well as in the McCoy's 5A cell culture medium supplemented with 15% FBS (to be used in the in vitro tests on cell cultures). It is our belief that the increased degree of biomimicry of the latter could foster a more truthful interrogation of the material degradation and bioactivity performance. After 7 days of immersion, the films thickness decreased by ~6.5% when immersed in SBF and by ~4% in the case of the tests carried out in the FBS supplemented McCoy's 5A culture medium. After 21 days of immersion a decrease of Bio-HA film thickness with ~14% and ~5% was recorded in SBF and FBS supplemented McCoy's 5A, respectively.

The AFM investigations revealed that radical morphological changes occurred on the surface of Bio-HA sputtered film immersed in SBF and FBS supplemented McCoy's 5A simulated physiological media (Figure 7). After 7 days of immersion in SBF on the surface of Bio-HA films, initially composed of a uniform bed of nano-grains (Figure 3c), was signaled the formation of spheroid conglomerates with a mean diameter of ~1–1.5 μm (Figure 7a). Such type of topography is specific to biomimetic HA layers formed in SBF solution [71,72] and demonstrates (according to ISO 23317:2014) the bioactivity of the Bio-HA sputtered film. For comparison, in Figure 7b a SEM image of a fully-developed biomimetic nano-crystalline HA layer is given, formed in 30 days in SBF on the surface of a bioactive silicate-based glass film [72]. The higher magnification AFM image (Figure 7c) evidenced that the micron-sized spheroid aggregates are constituted of randomly-dispersed small round particles (with sizes in the range of ~35–50 nm). A $R_{rms}$ value of ~4 nm was inferred. After 21 days of immersion in SBF, the AFM analyses (Figure 7e,g) indicated that the formation of the biomimetic calcium phosphate layer progressed, having at this time point a denser morphology (Figure 7g) and an overall lower $R_{rms} \approx 1$ nm.

In the case of tests performed in FBS supplemented McCoy's 5A solution, the biomineralization processes seem to take place at a slower pace. After 7 days of immersion the general appearance of the surface was found rather uneven. The presence of the spheroid aggregates is only hinted (Figure 7b), having a smoother morphology with respect to the SBF experiments (Figure 7a). On top of this covering blanket, smaller grains (~0.2–0.5 μm), randomly distributed, were revealed. Consequently, a larger $R_{rms}$ value of ~13 nm was inferred. After 21 days of immersion in the FBS supplemented

McCoy's 5A solution, the surface of the samples (Figure 7f,h) closely resembled the one observed in the case of films tested for 7 days in SBF (Figure 7a,c), further suggesting that biomineralization processes are slower in the FBS supplemented McCoy's 5A solution with respect to the ones occurring in SBF. Kindred observations were recently reported by Popa et al. [59], Vladescu et al. [11] and Vranceanu et al. [73] which showed that the (i) biomineralization processes (i.e., formation and structuring of biomimetic HA layers) takes place at a lower rate [59] and (ii) the degradation is significantly reduced [11,73] in media with a complex organic–inorganic composition (such as in the FBS-supplemented McCoy's 5A solution) with respect to those encountered in the inorganic SBF solution.

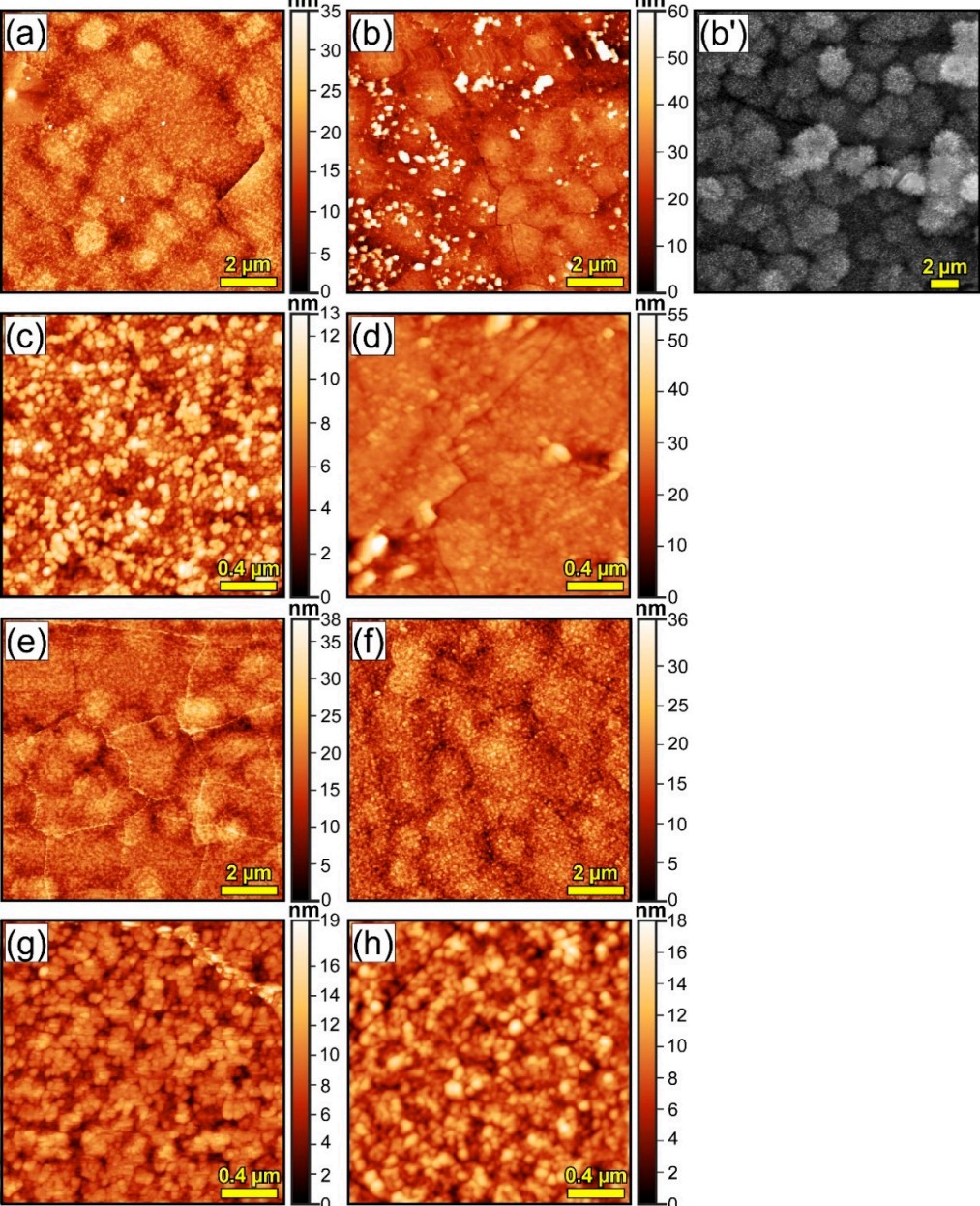

**Figure 7.** Characteristic AFM images of Bio-HA sputtered films tested in vitro in (**a,b,e,f**) SBF and (**c,d,g,h**) FBS supplemented McCoy's 5A simulated physiological solutions, for (**a–d**) 7 and (**e–f**) 21 days, acquired on scanning areas of: (**a,b,e,f**) 10 × 10 and (**c,d,g,h**) 2 × 2 $\mu m^2$, respectively. (**b′**) SEM images of a fully developed biomimetic nano-crystalline HA layer formed in 30 days in the SBF medium. Fig. 7b′ is adapted after Fig. 2a from Reference 72, with permission from Springer, 2019.

The FTIR comparative spectra of the Bio-HA films after immersion for 7 and 21 days in SBF and FBS supplemented McCoy's 5A media are presented in Figure 8. A strong diminution of carbonate bands intensity is noticed even after seven days, with the allure and intensity of the orthophosphate and hydroxyl bands basically unaltered. The decrease of the carbonate content of the Bio-HA films could be attributed to the weaker Ca–$CO_3$ bonds as compared to the Ca–$PO_4$ ones [74], which makes them more susceptible to be cleaved and released during the ion exchange processes with the surrounding medium [71]. No other remarkable structural changes have been revealed by FTIR measurements (Figure 8) up to 21 days of immersion, which suggests that the modification of the film surface revealed by AFM (Figure 7) is only superficial. However, one should note that the phase discrimination between the top biomimetic HA layer and the Bio-HA sputtered film underneath is impeded by their structural–chemical similarity (i.e., both being nano-structured and having akin compositions).

Overall, the results suggest a reduced rate of degradation accompanied by a good biomineralization capacity of the Bio-HA films, with the loss of mass being compensated to a certain extent by growth of new calcium phosphate layers in contact with the simulated physiological solutions.

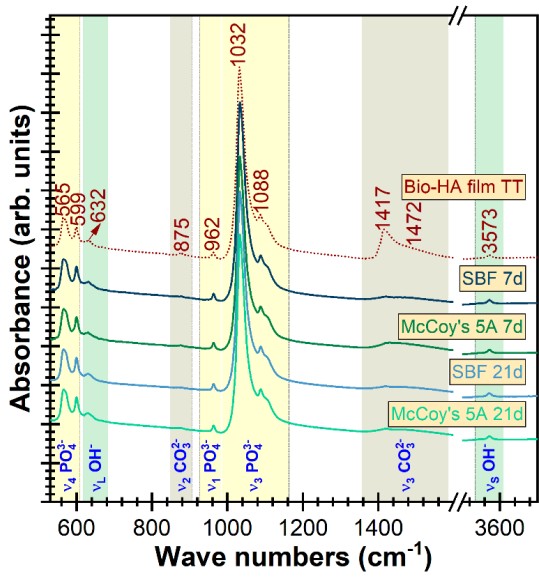

**Figure 8.** Comparative FTIR spectra of the Bio-HA sputtered film before and after in vitro testing for 7 and 21 days in SBF（simulated body fluid）and FBS (fetal bovine serum) supplemented McCoy's 5A simulated physiological solutions.

### 3.2.3. In Vitro Cytocompatibility Studies

To determine the effect of Bio-HA coating on cell viability and proliferation, we have carried out an MTS test at 1, 3, and 7 days after cell seeding. As a control, a borosilicate glass coverslip was used, which is a cell culture standard [75]. The results are represented in Figure 9 and they show an enhanced osteoblasts proliferation on the control up to 72 h. However, after 168 h, the Bio-HA coated substrate was found to encourage cells expansion, as indicated by a significant increase in MTS absorbance value. This result suggests a slower proliferation dynamic on Bio-HA at early time points, which is compensated after cells reach semi-confluency.

The adhesion of SaOs-2 cells on the Bio-HA film and the control surface has been investigated by fluorescence microscopy (Figure 10). The actin filaments were labelled in green to study the cytoskeleton organization, while the cell nuclei were represented in blue. As it can be seen from Figure 10a,b, after one day of culture, the cells have successfully attached on both substrates. However, in the case of the Bio-HA coating (Figure 10a), cells have extended their membrane

protrusions and changed their morphology to a more elongated phenotype. This difference in the cytoskeleton structure is more pronounced after three days of culture (Figure 10c,d). Cells grown on the control material (Figure 10d) elicit a more spread cell body compared to those on the Bio-HA substrate where they become rather elongated with multiple developed filipodia (Figure 10c). However, taking into consideration the increased cell number detected after 7 days by MTS (Figure 9) on the Bio-HA sputtered surface, we can conclude that this biomaterial is compatible with bone cell adhesion and expansion, which are key early steps important for osseointegration.

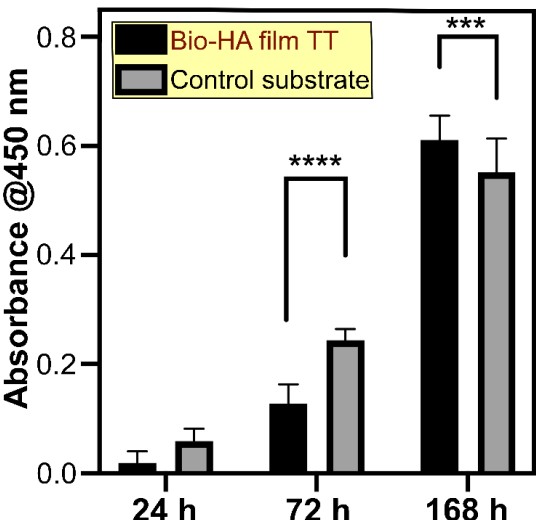

**Figure 9.** SaOs-2 cells proliferation on Bio-HA sputtered film and control substrate, after 1, 3, and 7 days of culture (***$p < 0.001$; ****$p < 0.0001$).

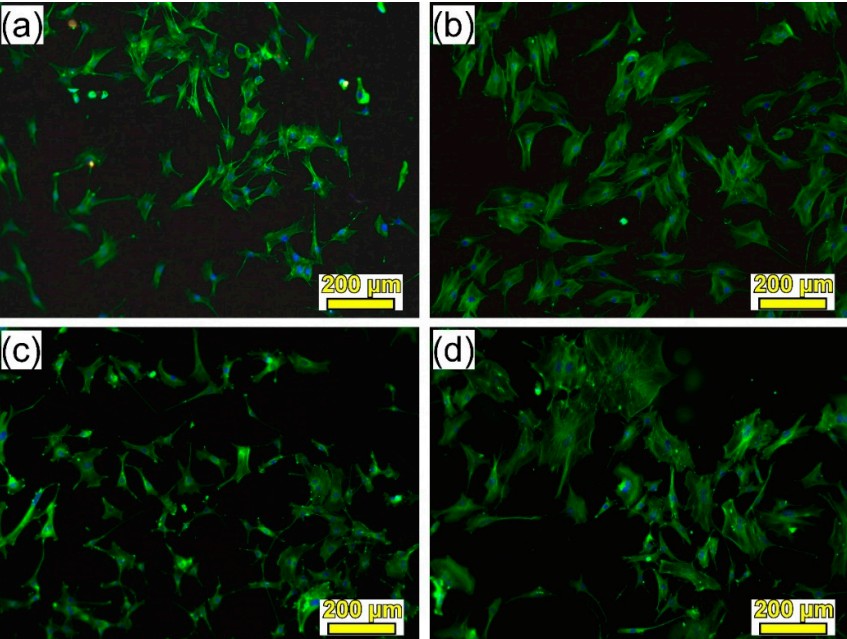

**Figure 10.** Immunofluorescence microscopy images of SaOs-2 cells grown on the surface of the Bio-HA coated 3D printed Ti6Al4V substrates (**a,c**) and control substrate (**b,d**), after one day (**a,b**) and three days (**c,d**) of seeding. The actin filaments are stained in green and the cell nuclei are labeled in blue. Scale bar = 200 µm.

## 4. Discussion

3D printing is a technique that resonates with medicine. The prospect of custom metal implants that can be produced rapidly for specific bone fractures is highly appealing. In general, metal printing techniques are not cheap and neither is prosthesis manufacturing. While 3D printing is not envisaged for mass production due to its high costs, it is an excellent option for manufacturing prototypes and complex shape items that cannot be obtained by conventional casting. 3D printed metal implants fall into this category, as they can be patient specific, fabricated in a low number for targeted needs and can support the high manufacturing cost. Among laser metal printing techniques, SLM and laser melting deposition (LMD) are the main options. In the case of cranial meshes, LMD is out of discussion, as it lacks the resolution for producing lines with submillimeter width. With the LMD experimental setup in our laboratory, we have not succeeded to produce lines thinner than 2.5 mm. This width increases in the case of multiple layer patterns, as material flows during deposition. In the case of SLM, this problem was non-existent, the minimum width of obtained lines being of ~200 μm, thus, allowing to obtain even cranial prostheses with intricate patterns.

However, one pending drawback of titanium-based implants, in spite of their excellent biocompatibility and mechanical performance, is their slow osseointegration. One solution to tackle this downside is the application of coatings of bioactive ceramics [34,76], an ever-growing and dynamic field of research if taking into account the recent statistics on yearly distribution of kindred scientific articles published in Reference 76. For this purpose, radio-frequency magnetron sputtering was selected as the deposition method of choice, due to its ability to generate high purity, dense, and adherent coatings on large area substrates (dependent only on the cathode gun size), making it scale-up and mass production ready. This study focused on sputtered hydroxyapatite films derived from biogenic sources (i.e., cortical bovine bones), which have both cost-efficient and biofunctionality valances. On the one hand, repurposing highly-abundant waste materials (such as animal bones, resulting in large quantities from the food industry) represents an inexpensive, sustainable, and environmentally-preferred alternative. On the other hand, biological apatite is substituted with low contents of cations (e.g., $Na^+$, $Mg^{2+}$) and anions (e.g., $CO_3^{2-}$, $F^-$, $SiO_4^{4-}$), each with specific biofunctional roles [17,18,65,66,74,76–78]. Such trace elements can be directly responsible for the modifications in crystallinity, degradation speed, biomineralization capacity, or mechanical performance of apatite [18,74,76].

Thereby, the current study is congruent with the current efforts taking place worldwide to delineate patient-customized implants capable to warrant a rapid osseointegration process. Furthermore, the importance of this study stems also from being the first attempt to assess RF-MS technique ability (to the best of our knowledge) to transfer biological apatite compounds in the form of thin films. The physical–chemical analysis results showed great promise, with the composition (presenting only slight deviations, to be expected in the case of a physical vapor deposition technique such as RF-MS) and structure (similar crystallite size, conservation of the B-type carbonatation) of the source biogenic apatite material being well reproduced into the crystallized sputtered films. The *c*-axis preferential orientation of the Bio-HA sputtered films could constitute a supplemental attribute to be further explored, since several studies have stressed the superiority of such textured structured with respect to the randomly oriented ones in terms of chemical stability in physiological environments [79], wear resistance [80], hardness and Young modulus [81], and bioactivity [81].

The in vitro degradation/bioactivity tests performed in the complex organic–inorganic McCoy's 5A FBS-supplemented medium (for a more rigorous evaluation), indicated on the one part the stability of Bio-HA film (a thickness decrease of only ~5% was recorded after three weeks of immersion) and the good biomineralization capacity highlighted by the formation of the characteristic spherulitic deposits of biomimetic calcium phosphates. The bone bonding mechanism of dense hydroxyapatite constructs (here including also magnetron sputtered films) is considered to consist of five consecutive reaction stages: (i) slight decrease of homeostatic pH at the interface between HA—biological environment; (ii) dissolution/precipitation processes culminating with the formation of the biomimetic carbonated HA; (iii) production of extracellular matrix by collagen and

adhesive proteins; (iv) concurrent mineralization of collagen fibrils and incorporation of carbonated HA within the new bone remodeling process; and (v) interlocking and strengthening of host bone—HA interface by mineralized collagen [82,83]. The first two stages can be evaluated in vitro in media with ion concentrations equivalent to those of human blood plasma [59,84], and currently, it is generally viewed that the in vitro capacity of a material to generate the formation of a biomimetic layer on its surface is a predictor of in vivo bone-bonding ability [84]. The partial dissolution process (in vivo mediated by extracellular fluids and/or phagocytosis by osteoclasts) of HA leads to a local concentration increase of $Ca^{2+}$ and $PO_4^{3-}$ ions which will upsurge even more the saturation of the micro-environment (already saturated with respect to calcium and phosphate) and trigger the nucleation and precipitation of apatitic crystals, which will incorporate in time carbonate, hydroxyl, and various cations in its further crystallization and structuring process [85,86]. Thereby, a bioactive material will always need to dissolve slightly in order to induce the formation of biological apatite which will facilitate the chemical bond with the bone [87]. A too fast or extreme degradation could lead to the loss of the osteogenic coating before fulfilling its role. However, in our case, after 21 days of immersion in the simulated body media solutions Bio-HA film experienced only a minor thickness reduction (i.e., of ~14% and 5%, in SBF and SBF and FBS-supplemented McCoy's 5A, respectively). The formation of biomimetic calcium phosphates layers (whose nucleation already started at 7 days, and fully developed at 21 days), shown for the Bio-HA RF-MS film, in both SBF and FBS-supplemented McCoy's 5A, already started to compensate for any further implant coating mass loss, and will hinder its further degradation. Thereby, it is expected for the Bio-HA films to preserve one of its designed roles (i.e., buffer against the release of metallic ions of the metallic substrate in the internal environment) until the bone regeneration process (typically taking 4–6 weeks) is complete [88].

The biomineralization processes were suggested to be slower in the medium with increased biomimicry with respect to purely inorganic SBF solution. This could be related to the unnatural high pH (i.e., ~8) attained typically by the SBF solution with respect to the FBS-supplemented McCoy's 5A (which had a constant homeostatic pH of 7.3–7.4 throughout the testing period) and/or to the adsorption of proteins from the medium [59] on the surface of the Bio-HA film, forming a capping layer which slightly retards the biomineralization processes. The development at a slower pace of HA in complex organic–inorganic testing media could be also connected with (i) the inhibitory effect of amino-acids which are known to chelate $Ca^{2+}$ and $PO_4^{3-}$ ions or bind to nuclei of calcium phosphate [89] and (ii) the proficiency of testing medium proteins to increase the free energy of nucleation and thereby, to reduce the rate of calcium phosphate nucleation by decreasing the supersaturation level [90]. However, since there is no interaction of an implant with purely inorganic medium such as SBF, it is our belief that only tests performed in solutions with an increased biomimicry degree could provide accurate gauges of their real in situ performances.

Furthermore, the cytocompatibility tests indicated that besides their bioactive behavior, the films were also non-toxic and stimulated the proliferation of osteoblast-like cells. Roughness was long advertised to play an important role on the biological response of implants [91,92]. Since the Bio-HA sputtered films were rather smooth ($R_{rms} \approx 15$ nm) and uniform, the in vitro biological response (i.e., bioactivity, cell shape modifications, and their accelerated proliferation rate) could be associated prominently to the chemical and structural nature of the biological HA coatings.

Overall the physico–chemical properties and preliminary biological behavior of the sputtered Bio-HA films hold promise, and thus encourage us to further and more insightfully assess their potential from mechanical and biofunctional points of view.

## 5. Conclusions

3D cranial meshes of Ti6Al4V were successfully manufactured by SLM. The microstructure of Ti6Al4V deposited by SLM was martensitic $\alpha'$. The irradiation parameters were optimized in order to obtain shapes without defects, such as pores or cracks and with homogeneous distribution of alloy elements in their volume.

In a second technological step the implants produced by SLM were coated by radio-frequency magnetron sputtering with a ~600 nm layer of hydroxyapatite derived from biogenic sustainable resources (i.e., calcined cortical bovine bones), which could constitute a cost-efficient, environmental-friendly, and highly promising biofunctional solution. Remarkably, the composition and structure of the biological apatite material were well-replicated into the smooth ($R_{rms} \approx 15$ nm) and uniform sputtered post-deposition crystallized films. Consequently, the Bio-HA films consisted of a single B-type carbonated hydroxyapatite phase with traces of $Na^+$ and $Mg^{2+}$, having a *c*-axis preferential texturing, similar to the bone mineral phase.

The Bio-HA sputtered films elicited promising biological performances:

1.  Chemical stability in simulated physiological solutions (after 21 days the film thickness was reduced by ~14% in SBF and ~5% in FBS-supplemented McCoy's 5A medium);
2.  Biomineralization capacity (Bio-HA generated the formation of spherulitic deposits of biomimetic calcium phosphates in contact with both types of simulated physiological solutions);
3.  Absent cytotoxicity;
4.  Optimal osteoblast cell proliferation and adhesion.

3D printing is now viewed as a highly appealing approach for the manufacturing of personalized implants. The most important functional request of such devices should be their capacity to stimulate fast bone regeneration. Thereby, the materials of choice would be bioceramics that highly resemble the composition and structure of the mineral bone phase. Unfortunately, such stand-alone bioceramics lack suitable mechanical properties risking implant failure. Furthermore, they are difficult to be 3D printed. Therefore, the compromise solution is the fabrication of customized implants by 3D printing from metallic materials, followed by their coating with bioactive ceramic thin films, closely mimicking the fine and intricate features of the metallic biomedical device. In this study we aimed to show that RF-MS could be a candidate deposition technique for the coating of such medical devices with thin films of biological apatite, on the one hand due to its now demonstrated ability to preserve the physico–chemical properties of the source material, and on the other hand due to its recognized capability to synthesize, in a controlled manner, dense and adherent thin films on large area substrates.

**Supplementary Materials:** The following are available online at www.mdpi.com/xxx/s1, Figure S1: (a) Optical microscopy image and (b) SEM micrograph of the Ti6Al4V sample produced by the SLM method. (c) A typical EDXS spectrum recorded for the Ti6Al4V sample 3D printed by SLM. Figure S2: (a) A typical EDXS spectrum recorded for the Ti6Al4V sample 3D printed by SLM. Compositional EDXS maps for (b) all constitutive elements (overlayed), (c) Ti, (d) Al and (e) V, collected for the area highlighted in the microscopic field presented in (Figure S1b).

**Author Contributions:** Conceptualization, A.C.P., G.E.S., F.N.O. and L.D.; SLM manufacturing, N.P. and N.M.; Metallographic investigations and writing—original draft preparation, D.C. and A.C.P.; SE, SEM, XRD, FTIR, AFM, EDXS, and in vitro degradation/bioactivity testing, G.P.P., L.M.B., A.C.G., A.C.P., and G.E.S.; In vitro cytocompatibility testing, S. O.; writing—review and editing, L.D., G.E.S., and A.C.P.

**Funding:** INFLPR authors acknowledge the funding of the Core Program contract no. 16N/2019 and of the Romanian Ministry of Research and Innovation in the framework of the projects PN-III-P2-2.1-PED-2016-1309 (PED241/2017), PN-III-P1-1.1-TE-2016-2015 (TE136/2018), and PN-III-P1-1.1-PD-2016-1568 (PD6/2018). NIMP authors are thankful for the support of PCCDI—UEFISCDI, in the framework of project PN-III-P1-1.2-PCCDI-2017-0062 (contract no. 58PCCDI/2018)/component project no. 2 and of Core Program (contract no. 21N/2019). This work was also supported by a grant of the Romanian Ministry of Research and Innovation, PCCDI—UEFISCDI, project number PN-III-P1-1.2-PCCDI-2017-0224 (contract no. 77PCCDI/2018)—DigiTech, within PNCDI III.

**Acknowledgements:** In vitro studies were conducted at the Biology Lab of CETAL Department.

**Conflicts of Interest:** The authors declare no conflict of interest.

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
