# Peer review of "Animal Origin Bioactive Hydroxyapatite Thin Films Synthesized by RF-Magnetron Sputtering on 3D Printed Cranial Implants"

_metals, doi:10.3390/met9121332_

Round 1

Reviewer 1 Report

The paper is very interesting and could have a high impact on the Readers of Metals. I would suggest a couple of small adjustmnents as follows: 

Hardness tests are mentioned in the Results but not in the Materials and Methods section, please include it; Layer thickness is section 2.1.1 is not mentioned but it would be interesting to include it in the 3D printing parameters list; There is a high number of experimental results reported in the paper, however Figure 2 could be omitted since it does not provide additional information. 

Author Response

Letter of response to Reviewer 1

The authors are grateful to this Reviewer for her/his careful suggestions. Our responses are highlighted in green and are listed in the order they appear in the report.

Comment 1: Hardness tests are mentioned in the Results but not in the Materials and Methods section, please include it.

Response 1: Thank you for this observation! We have included now these experimental details in the Materials and Methods section (page 4, lines 186 – 189). The new text reads:

The micro-hardness of the printed material was evaluated Falcon 500 mechanical testing machine (Innovatest, Netherlands) equipped with a Vickers indenter. Ten Vickers indentations have been made on the surface of the specimens using a load of 0.5 kgf/mm2. The reported micro-hardness value represents the average of the ten independent values ± standard deviation.

Comment 2: Layer thickness is section 2.1.1 is not mentioned but it would be interesting to include it in the 3D printing parameters list.

Response 2: We thank the reviewer for noticing this omission in the Materials and Methods section. We have mentioned now this information in section 2.1.1 (page 3, lines 163 – 164). The new text reads:

For the manufacturing of a 3 mm thick cranial mesh prosthesis a number of 100 successive layers were needed, corresponding to an individual layer thickness of 30 µm.

Comment 3: There is a high number of experimental results reported in the paper, however Figure 2 could be omitted since it does not provide additional information.

Response 3: We adopted the modification, and removed Figure 2 from the manuscript, split it in two new separate figures (Fig. S1 and Fig S2) and transferred to the Supplemental Information file. All Figures were renumbered accordingly. 

Reviewer 2 Report

Authors presented interesting research, but the manuscript requires the correction before publication. 1) On the page 9, lines 333 and 334 the references to Fig. 4 are wrong. Here Fig. 2 is presented. 2) Main question is why did decreased the thickness of Bio-HA coating in vitro tests in simulated physiological solutions? The experiment must show the apatite-forming ability of implant materials, but in your case the degradation of the coating is shown. How it may be explained?

Author Response

Letter of response to Reviewer 2

The authors are grateful to this Reviewer for her/his suggestions. Our responses are highlighted in cyan, and are listed in the order they appear in the report.

Comment 1: On the page 9, lines 333 and 334 the references to Fig. 4 are wrong. Here Fig. 2 is presented.

Response 1: Thank you for this observation! We have corrected now this error (page 15, lines 384 – 386).

Comment 2: Main question is why did decreased the thickness of Bio-HA coating in vitro tests in simulated physiological solutions? The experiment must show the apatite-forming ability of implant materials, but in your case the degradation of the coating is shown. How it may be explained?

Response 2: We have clarified this aspect now in the Discussion section (pages 15 – 16, lines 714 - 754). It now reads as:

The in vitro degradation/bioactivity tests performed in the complex organic-inorganic McCoy’s 5A FBS-supplemented medium (for a more rigorous evaluation), indicated on the one part the stability of Bio-HA film (~thickness decrease of only ~5% was recorded after 3 weeks of immersion) and the good biomineralization capacity highlighted by the formation of the characteristic spherulitic deposits of biomimetic calcium phosphates. The bone bonding mechanism of dense hydroxyapatite constructs (here including also magnetron sputtered films) is considered to consist of five consecutive reaction stages: (i) slight decrease of homeostatic pH at the interface between HA – biological environment; (ii) dissolution/precipitation processes culminating with the formation of the biomimetic carbonated HA; (iii) production of extracellular matrix by collagen and adhesive proteins; (iv) concurrent mineralization of collagen fibrils and incorporation of carbonated HA within the new bone remodeling process; and (v) interlocking and strengthening of host bone – HA interface by mineralized collagen [82,83]. The first two stages can be evaluated in vitro in media with ion concentrations equivalent to those of human blood plasma [59,84], and currently, it is generally viewed that the in vitro capacity of a material to generate the formation of a biomimetic layer on its surface is a predictor of in vivo bone-bonding ability [84]. The partial dissolution process (in vivo mediated by extracellular fluids and/or phagocytosis by osteoclasts) of HA leads to a local concentration increase of  and ions which will upsurge even more the saturation of the micro-environment (already saturated with respect to calcium and phosphate) and trigger the nucleation and precipitation of apatitic crystals, which will incorporate in time carbonate, hydroxyl, and various cations in its further crystallization and structuring process [85,86]. Thereby, a bioactive material will always need to dissolve slightly in order to induce the formation of biological apatite which will facilitate the chemical bond with bone [87]. A too fast or extreme degradation could lead the loss of the osteogenic coating before fulfilling its role. However, in our case, after 21 days of immersion in the simulated body media solutions Bio-HA film experienced only a minor thickness reduction (i.e., of ~14% and 5%, in SBF and SBF and FBS-supplemented McCoy’s 5A, respectively). The formation of biomimetic calcium phosphates layers (whose nucleation already started at 7 days, and fully developed at 21 days), shown for the Bio-HA RF-MS film, in both SBF and FBS-supplemented McCoy’s 5A, already started to compensate for any further implant coating mass loss, and will hinder its further degradation. Thereby, it is expected for the Bio-HA films to preserve one of its designed roles (i.e., buffer against the release of metallic ions of the metallic substrate in the internal environment) until the bone regeneration process (typically taking 4-to-6 weeks) is complete [88].

Round 2

Reviewer 2 Report

Manuscript is suitable for publication.